# Spatial-Temporal Variations in of Soil Conservation Service and Its Influencing Factors under the Background of Ecological Engineering in the Taihang Mountain Area, China

**DOI:** 10.3390/ijerph20043427

**Published:** 2023-02-15

**Authors:** Feng Wang, Jintong Liu, Tonggang Fu, Hui Gao, Fei Qi

**Affiliations:** 1Key Laboratory of Agricultural Water Resources, Hebei Laboratory of Agricultural Water-Saving, Hebei Key Laboratory of Soil Ecology, Center for Agricultural Resources Research, Institute of Genetics and Developmental Biology, Chinese Academy of Sciences, 286 Huaizhong Road, Shijiazhuang 050021, China; 2University of Chinese Academy of Sciences, Beijing 100049, China

**Keywords:** NDVI, slope, soil types, InVEST, geographical detector model (GDM)

## Abstract

Soil conservation (SC) plays an important role in maintaining regional land productivity and sustainable development. Ecological engineering (EE) is being implemented in different countries to effectively alleviate the damage to the ecological environment and effectively protect soil and food security. It is important to determine whether or not the SC capacity becomes stronger after the implementation of EE and whether or not EE has a notable impact on SC in different altitude zones. The exploration of the influencing mechanism and identification of the dominate influencing factors in different geographical regions needs to be improved. In this study, the soil conservation services (SCSs) from 1980 to 2020 in the Taihang Mountain area was assessed using the integrated valuation of ecosystem services and trade-offs (InVEST) model, and the spatial and temporal distributions and influencing factors were explored. The results showed the following: (1) the average SCSs exhibited an increasing trend from 1980 to 2020 on the whole, and the rate of increase reached 50.53% during the 41-year period. The rate of increase of the SCSs varied in the different EE implementation regions, and it was significantly higher than that of the entire study area. (2) The spatial distribution of the SCSs was highly heterogeneous, and the high SCS value areas were coincident with the high-altitude areas where forest and grassland occupied a large proportion. The low value areas were mainly located in the hilly zone or some of the basin regions where the proportion of construction land was relatively high. (3) The distribution pattern of the SCSs was the result of multiple factors. The EE intensity had the strongest explanatory power for the SCSs in the hilly zone, explaining 34.63%. The slope was the most critical factor affecting the SCSs in the mid-mountain and sub-alpine zones. The slope and normalized difference vegetation index (NDVI) had the greatest interactions with the other factors in the three altitude zones, especially in the high-altitude regions. The quantitative analysis of the SCSs and the influences of EE and natural factors on the SCSs revealed the heterogeneity in the mountainous areas. These results also provide a scientific basis for the reasonable implementation of EE and sustainable management of SCSs in the Taihang Mountain area.

## 1. Introduction

Soil conservation services (SCSs) are one of the critical ecosystem regulation services. SCSs have played an essential role in preventing a major global environmental problem and maintaining regional ecological safety [1,2,3]. Moreover, SCSs significantly influence the United Nations (UN) Sustainable Development Goals, for instance, halting and reversing land degradation. With urbanization and industrialization, SCSs have degraded and become unsustainable globally [4]. Ecological engineering (EE) has developed and increased rapidly worldwide in the last few decades [5,6], which could solve this problem to a certain extent. EE is defined as the design of sustainable ecosystems that integrate human society with the natural environment [7], and it could have a significant impact on SCSs [8]. Therefore, it is of great significance to study SCSs under the effect of EE.

With the implementation of EE, researchers have explored whether SCSs have been effectively improved. For instance, some studies of different implementation nodes of EE have found that the vegetation restoration resulting from the Grain for Green Project (GGP) played a positive role in enhancing SCSs [9,10,11]. In contrast, others reported a decreasing trend [12]. The spatial heterogeneity of SCSs under the influence of different forms of EE is remarkable [13,14]. SCSs have complex links with the ecological process of EE [15]. To our knowledge, few investigations have focused on the spatial distribution and temporal variations of SCSs combined with accurately quantifying EE.

The influencing factors of SCSs consist of natural and human factors (e.g., EE). Some authors have discussed the influences of natural factors such as vegetation type, precipitation, and geomorphology factors on SCSs [13,16,17]. While the influencing factors involved in their studies have some deficiencies. EE, one of the most crucial human factors, has not been taken into account in their studies. A previous study demonstrated that appropriate EE measures had a positive effect on SCSs [18]. EE can directly affect the distribution of SCSs by changing the land use types [19,20]. In addition, researchers have investigated other influencing forms of EE, which consist of types such as the afforestation area, vegetation restoration, and relative policies [13,21,22]. For instance, the vital soil conservation (SC) function areas were generally consistent with high vegetation coverage areas [23]. However, vegetation restoration generally had a threshold effect in improving SC capacity [24]. The improvement of SC capacity resulting from EE was accompanied by a significant decrease in soil water content [25]. Therefore, clarifying the critical areas of EE was crucial to maximizing the effect of EE and guaranteeing the balance of SC and other ecosystem services.

Most mountainous regions have high SCS values on the national and global scales [26,27]. Different from those plain areas, mountains have significant environmental gradients, and the interactions between EE and ecological processes could strongly influence SCSs in mountain areas [28]. Therefore, it is more challenging to analyze the influencing mechanism of SCSs in these regions. It can be concluded that the vertical gradient characteristic is a crucial factor in the ecological process that contributes to the heterogeneity of SC [29,30]. The relationships between EE and SCSs along these altitude zones in mountainous areas mark these areas as being of particular interest to researchers.

The Taihang Mountains, a typical mountainous area in northern China, are the ecological barrier of the Beijing-Tianjin-Hebei and are the production base of the North China Plain. The SCSs are vital to this arid and semi-arid area [31], and the hilly zone in the southwest of the study area is nationally critical for soil and water conservation [32]. Some authors found SCSs in these high-altitude areas were higher than that in low-altitude areas [29,33]. The ecological environment is relatively fragile, and the intensity of human activities has been high historically, resulting in severe soil and water loss [29,34]. In order to solve ecological problems, improving SC function, the Three-North Shelterbelt Project, Beijing-Tianjin Sandstorm Source Control Project, and GGP have been successively carried out over the last few decades in this area [35]. What are the effects of these ecological projects? Most previous studies involving SCSs still lack consideration of the EE factor [29,30,33], especially in diverse terrain conditions. Therefore, regional sustainable development will be influenced by SCSs change and implementation of EE.

In this study, the SCS values in 1980, 1990, 2000, 2010, and 2020 were calculated using the integrated valuation of ecosystem services and trade-offs (InVEST) model. This study analyzed the capacity of the SC, clarified the different EE implementation areas. The three altitude zones (hilly, mid-mountain, and sub-alpine) are considered to evaluate the variations of SCSs using the geographic detector model (GDM). The results of this study provide scientific guidance for the management of SC ecosystem services and regional ecological environment improvement in the Taihang Mountain area.

## 2. Materials and Methods

### 2.1. Study Area

The Taihang Mountains are located in northern China (110°14′–114°33′ E, 34°34′–40°43′ N) and extend in the southwest-northeast direction (Figure 1a). The Taihang Mountain area extends across a total of four provinces (Beijing, Hebei, Shanxi, and Henan) and 101 counties. It is an critical transition zone between the Loess Plateau and the North China Plain. The average elevation in the Taihang Mountains is 1000–1500 m, which decreases from the northwest to the southeast. The highest elevation (3058 m) is located in the territory of Xinzhou City (Figure 1b).

The Taihang Mountains have a typical East Asian Monsoon climate, with an average temperature of 11.4 °C and an average annual precipitation of 456.5 mm. The rainfall exhibits high seasonal variability, with the highest precipitation in July (130 mm) and the lowest precipitation in December (10 mm) [29]. The distribution of the vegetation is significantly affected by the precipitation, temperature, topography, elevation, and slope, as well as the interactions among these environmental factors [31].

The soils are always thin in the Taihang Mountains, with an average thickness of 35 cm [36]. The soils are mainly developed from limestone in the northern and southern regions and from gneiss in the central region [37]. There are 26 soil types, the most common of which are Cambisols and the most uncommon of which are Podzoluvisols. The soil types have been described in detail by Fu et al. [38].

### 2.2. Data Sources and Pretreatment

The following data were used in this study (Table 1 and Figure 2). The precipitation data used in this study were for 1980, 1990, 2000, 2010, and 2020, obtained from 88 meteorological stations in 101 counties, approximately occupying 90%. Land-use/land-cover (LULC) data for 1980, 1990, 2000, 2010, and 2020 we used were obtained by visual interpretation using TM images and Landsat 8 images. They were divided into six categories (Table 1) and provided by the Resources and Environment Science Data Center at the Chinese Academy of Sciences. The topographic map of the study area was constructed using a digital elevation model (DEM), processed by mosaicking to a new raster. The soil data were derived from China’s second national soil survey, with a proportional scale of 1:1 million. In order to improve the calculation accuracy, the DEM, soil, vegetation, and LULC data were resampled to a spatial resolution of 100 m × 100 m using the ArcGIS 10.2 software.

### 2.3. InVEST Model

The InVEST model [39] (i.e., InVEST 3.9.0) is a spatially explicit model that operates at the cell size of the DEM. It is a suite of models used to map and determine the values of natural goods and services that sustain and fulfill human life. Therefore, we applied this model to the Taihang Mountain area to estimate the annual SC. The model first computes the amount of eroded sediment and then the sediment delivery ratio (SDR), which is the proportion of lost soil that actually reaches the catchment outlet. The soil loss module estimates the annual soil loss of each sub-watershed according to the revised universal soil loss equation (RUSLE) model algorithm [40]. This approach was selected since it requires a minimal number of parameters, uses globally available data, and is spatially explicit.

The soil loss module of the InVEST SDR determines the long term mean annual soil loss rate of each pixel using five major RUSLE factors, which is expressed in Equation (1) [39,40,41].
*A* = *R* × *K* × *LS* × *C* × *P*,(1)
where *A* is the average annual soil loss rate (t·hm^−2^·yr^−1^), *R* is the rainfall erosivity factor [(MJ·mm)/(hm^2^·h·a)], *K* is the soil erodibility factor [(t·hm^2^·h)/(MJ·hm^2^·mm)], and *LS*, *C,* and *P* are the topographic, crop management, and erosion control practice factors, respectively.

#### 2.3.1. Rainfall Erosivity Factor

The rainfall erosivity factor (R) accounts for contribution of the erosive power of a certain rainfall to the overall erosion [42]. Previous studies in the Taihang Mountain area used the classical calculation formula to drive the R [35] although the rainfall intensity data are the most important variable for correctly determining the rainfall erosivity. At present, the *R* is usually estimated using the annual and monthly rainfall in China. In this study, the obtained rainfall and rainfall erosivity data were monthly data, and the monthly values were converted to the annual time scale. *R* (Figure 3) for 1980, 1990, 2000, 2010, and 2020 was calculated using Equation (2) applied by Ma [43] in this study.
(2)$$R=1.2157\times \sum_{i=1}^{12}10^{1.5(\mathrm{lg}\frac{P_{i}^{2}}{P}-0.8188)},$$
where *P_i_* is the total rainfall (mm) in month *i*, and *P* is the total annual rainfall (mm). The ANUSPLIN interpolation method [44] and raster resampling were applied in ArcGIS to obtain an accurate spatial distribution map of the rainfall erosivity *R*, with a 100 m accuracy.

#### 2.3.2. Soil Erodibility Factor

Soil erodibility factor (*K*) mainly expresses the soil properties. The *K* value reflects the degree of difficulty with which soil can be separated, eroded, and transported by rainfall. The *K* value is a comprehensive index of the resistance to water erosion. The higher the *K* value is, the weaker the soil’s resistance to water erosion is and the more susceptible it is to erosion [45]. The value of *K* ranges from <0.1 for the least erodible soils to approximately 1.0 for the worst possible case (most erodible soils). In this study, *K* was calculated based on the erosion-productivity impact calculator (EPIC) model [46] (soil texture and soil organic carbon content, Equation (3)), and the results were corrected according to the method of Zhang [47]. Therefore, *K* (Figure 4) was calculated using Equation (5) and the rater calculator tool in ArcGIS 10.2. Based on the *K* value assignment, conversion into maps with 100 m grid cells was undertaken using the soil physical-chemical property data. The *K* values in this study ranged from 0.05 to 0.28.
(3)$$\begin{array}{l} K_{EPIC}=\left\{0.2+0.3\times \exp\left[-0.0256SAN\left(1-\frac{SIL}{100}\right)\right]\right\}\times {\left(\frac{SIL}{CAL+SIL}\right)}^{0.3}\times \\ \left[1-0.25\times \frac{C}{C+\exp\left(3.72-2.95C\right)}\right]\times [1-0.7\times \frac{SN_{1}}{SN_{1}+\exp\left(22.9SN_{1}-5.51\right)}], \end{array}$$
(4)$$SN_{1}=1-\frac{SAN}{100},$$
(5)$$K=-0.01383+0.51575K_{EPIC},$$
where *SAN* (particle size: 0.05–2 mm), *CAL* (particle size: <0.002 mm), *SIL* (particle size: 0.002–0.05 mm), and *C* are the mass-based percentages of sand, silt, clay, and organic carbon, respectively.

### 2.4. Quantification of the Intensity of Ecological Engineering

The human activity intensity of the land surface (HAILS) method proposed by Xu Yong et al. [48] was used to quantify the intensity of the ecological engineering (IEE). Ecological engineering is a form of human activity [49]. Although some scholars have used various factors to describe the human activity intensity [50], these factors are highly correlated with land use, and their information can be well represented by land use types. Thus, in this study, the IEE was estimated based on land use data. The formula for calculating the IEE is as follows [48]:(6)$$IEE=\frac{S_{ISE}}{S}\times 100\%$$
(7)$$S_{ISE}=\sum_{i=1}^{n}SL_{i}\times CI_{i}$$
where *IEE* is the intensity of the ecological engineering; *S_ISE_* is the area of the imperious surface equivalent; *S* is the total land area of a certain region; *SL_i_* is the area of land use type *i*; n is the number of land use types; and *CI_i_* is the conversion coefficient of type *i* for the imperious surface equivalent. The *CI_i_* value of each land use type was determined according to the research results of Xu Yong et al. [48].

### 2.5. Geographical Detector Model

The geographical detector model (GDM) is a statistical method that reveals the spatial effects of the explanatory variables on the interpreted variables and identifies the interactions between the driving factors. It is a model that measures the driving force in the spatial differentiation mechanism, and it includes risk detection, factor detection, ecological detection, and interaction detection [51]. The detection contributes to clarification of the statistical significance of the independent variable and the explanatory power of the dependent variable. When faced with multiple influencing factors, the GDM can better overcome the problem of complicated calculations. Interaction detection is used to identify the interactions between the different factors, that is, to evaluate the influence of factors *X*_1_ and *X*_2_ on the explanatory power of the dependent variable. The GDM has been widely used in the field of ecology [16,52]. Based on the collinear immunity of the geographic detectors to the independent variables, the GDM was applied to quantitatively analyze the contribution rates of six driving factors to the SCSs as follows:(8)$$q=1-\sum_{h=1}^{L}N_{h}\sigma_{h}^{2}/N\sigma^{2},$$
where *h* = 1, …, *L* is the stratification of *Y* or *X*; *N_h_* and *N* are the numbers of units in layer *h* and the entire area, respectively; $\sigma_{h}^{2}$ and $\sigma^{2}$ are the variances of layer *h* and the entire area, respectively; and q∈[0,1]. The larger the *q* value is, the greater the contribution of the driving factor to the SCSs is. In this study, factor detection was used to explore the individual effects of the influencing factors on the SCSs, and interaction detection was used to evaluate the joint effects of the different factors on the SCSs when they interacted.

## 3. Results

### 3.1. Model Validation

It is an effective method to adjust and calibrate this model according to the knowledge of the hydrologic regime in the study area. The sed-export data simulated using the InVEST model should be compared to any observed sediment loading at the outlet of the watershed. The *IC*_0_ and *k_b_* values were important input parameters, and we set these values to 0.5 and 2 based on repeated analysis and comparison.

The long-term average annual sediment yield at five gauge stations in the studied basin (Figure 5), namely, Shixiali, Zhongtangmei, Xiaojue, Pingshan, and Guantai stations, were recorded as 4.97 t hm^−2^ yr^−1^, 4.37 t hm^−2^ yr^−1^, 5.63 t hm^−2^ yr^−1^, 6.8 t hm^−2^ yr^−1^, and 5.38 t hm^−2^ yr^−1^, respectively (the time period of the statistical data: from its establishment to 2000). For instance, the value measured at Zhongtangmei station was 4.37 t hm^−2^ yr^−1^, while the average sediment export value calculated using the InVEST model was 4.03 t hm^−2^ yr^−1^. The fluctuation in the simulated result was within the controllable scope, except for at Pingshan station. Thus, it was possible to confirm that the parameters of the model used in this study were able to reliably simulate the general sediment export, and the model results for the study area were reasonable.

### 3.2. Temporal Changes in Ecological Engineering and Soil Conservation

The results obtained using the InVEST model are shown in Figure 6. The average annual SCSs were 208.23 t hm^−2^ yr^−1^, 210.18 t hm^−2^ yr^−1^, 344.06 t hm^−2^ yr^−1^, 267.63 t hm^−2^ yr^−1^, and 313.46 t hm^−2^ yr^−1^ in 1980, 1990, 2000, 2010, and 2020, respectively (Figure 7a). The total amount of SC increased from 2.85 × 10^9^ t in 1980 to 4.29 × 10^9^ t in 2020, and the SCSs increased by 50.53% during the 41-year period.

As shown in Figure 7a, the IEE was the lowest in the initial implementation phase (1980), with a value of 15.6, and the value steadily increased beginning in 1980. The GGP had been implemented nationwide in 2002. Therefore, the IEE exhibited the largest increase in 2010. The IEE was greater than 19 in 2020, which was significantly higher than that of 1980. During the 41-year period from 1980 to 2020, the variation tendency of the IEE was always positive.

The SC function was further promoted after the implementation of EE, and it exhibited differences in the following EE areas (Figure 7b). For instance, the average amount of SC in the returning farmland to forestland (RFTF) area increased by 58.44% from 1980 to 2020, while the growth rates in the other EE areas, namely, returning farmland to grassland, returning grassland to forestland, returning farmland to wetland, and returning unused land to forestland and grassland, were 76.25%, 57.42%, 58.60%, and 105.44%, respectively, with an average value of 71.23% in all of the EE areas. The rates of increase of the SC in these areas during the 41-year period were significantly higher than the average value for the entire study area.

### 3.3. Spatial Distribution of Soil Conservation Services

The SC function in the Taihang Mountain area exhibited apparent spatial heterogeneity in the vertical and horizontal directions. In the horizontal direction, using the Hot Spot Analysis (Getis-Ord Gi∗) module embedded in the ArcGIS software, we found that the SCS hot spots were mostly clustered in the high-altitude areas in the eastern part of Xinzhou, southwestern part of Beijing, and southern part of Jincheng (Figure 8). In contrast, most of the cold spots were located in areas with flat terrain, exhibiting a conspicuous zonal distribution along the north–south direction, such as the eastern hilly zone (e.g., Xingtai County) and central basin (e.g., Changzhi Basin) in the Taihang Mountains. The SCS hot spots were always larger than the cold spots. Compared with the cold and hot spots, the non-significant areas accounted for a larger proportion, except for in 2000. The areas of the SCS cold and hot spots were the largest in 2000, occupying 50.43% (Table 2). Relatively, the area of the SCS cold and hot spots in 2020 was very small (13.8%), and the proportion of the non-significant areas reached the maximum (86.20%). This was probably due to the increase in precipitation in 2000, and the SC function tended to become more concentrated and divergent with increasing rainfall intensity.

Regarding the vertical direction, the SC function in the Taihang Mountain area varied with altitude. The average SCS value was 149.71 t hm^−2^ yr^−1^ in the hilly zone, calculated using the data for the sampling sites, which was lower than in the mid-mountain and sub-alpine zone (Figure 9). This was undoubtedly due to the fact that the SC function of the forest land was the strongest. The proportion of forest land in the hilly zone was 9.13%, which was significantly less than that of the construction land (16.31%). Human activities were mainly concentrated in the hilly zone. The SC function was stronger at higher elevations due to the higher proportion of forest land or grassland. The coverage rate of the forest land and grassland in the sub-alpine zone in the Taihang Mountain area was as high as 91.13% (Table 3). Therefore, the average SCS value was the highest (456.96 t hm^−2^ yr^−1^), which was 1.27 times and 3.05 times those of the mid-mountain and hilly zones, respectively.

### 3.4. Quantitative Identification of the Relationships between the Soil Conservation and Variables in the Different Altitude Zones

In order to clarify the driving force, in this study, we analyzed the relationship mechanism between the SCSs and their driving factors using the geographical detector model. The topography (slope or altitude), vegetation, climatic change (precipitation), and human activities were considered to be the main factors influencing the SCSs in many previous studies [23,29]. Therefore, six indexes (precipitation, normalized difference vegetation index (NDVI), slope, elevation, IEE, and soil type) were selected for quantitative analysis.

#### 3.4.1. Attribution Analysis of Single Dominant Factors

The relationships between the SCSs and variables were found to have different dominant factors with varying explanatory powers in the different altitude zones (Figure 10). In the hilly zone, the IEE (i.e., the human factor used in this study) was the dominant factor affecting the spatial heterogeneity of the SCSs, and its explanatory power was 34.63% (Figure 9a). The explanatory powers of all of the natural factors did not exhibit notable differences, and their q value were as follows: soil types > NDVI > slope > precipitation > elevation. In the mid-mountain zone, the IEE was no longer the dominant factor, and its explanatory power for the SCSs was only 18.71%. The slope had the most significant explanatory power (38.74%) for the SCSs, and the q values of the PRE and NDVI were greater than 0.14. In the sub-alpine zone, the IEE had the lowest explanatory power of only 3.11%. Compared with the human factor, the influence of the natural factors was more crucial to the SCSs, with a cumulative q value of 0.97. Of the natural factors, the slope and NDVI were the dominant influencing factors, and their total q value was greater than 0.65. Moreover, this was accompanied by thicker vegetation in certain regions, contributing to a strong SC capacity in areas with higher elevations and steeper slopes. In most of these regions, the proportion of woodland and grassland was significantly higher than that of the other land use types. Therefore, the slope made a larger contribution to the SCSs in the mid-mountain and sub-alpine zones. In general, the explanatory power of the IEE for the SCSs gradually decreased with increasing altitude, while the natural factors became increasingly significant with increasing altitude.

#### 3.4.2. Quantitative Attribution of Interaction Factors

The results of the interaction detection using the GDM indicate that the explanatory power of the interaction between any two factors was stronger than the explanatory powers of the single factors on the spatial distribution of the SCSs. The types of interactions between the factors were mainly manifested nonlinear enhancement and double factor enhancement. These interactions demonstrate that the spatial pattern of the SC in the Taihang Mountain area was influenced by multiple factors. In the hilly zone, the SCSs were mostly affected by the IEE (Figure 11). The top three interactions in terms of the explanatory power for the SCSs were all interactions between the IEE and other factors. In the mid-mountain zone, the interactions between the slope, NDVI, and the other four factors were the largest, and the q values of these interactions were all greater than 0.13. In the sub-alpine zone, the slope was one of the dominant interaction factors affecting the SCSs, and the interaction between the slope and NDVI had the greatest explanatory power. Overall, the slope and NDVI had the greatest interactions with the other factors in the three altitude zones, especially in the high-altitude zones. The multi-factor interactions significantly enhanced the explanatory power for the SCSs in the study area.

## 4. Discussion

### 4.1. Ecological Engineering and Its Impact on Land Use Change

Since the 1980s, soil and water conservation projects have been implemented in the Taihang Mountain area, including the construction of terraces and check dams, the afforestation project of the Taihang Mountains (APTM), and the GGP. The cumulative area of the APTM reached 4.18 million hectares at the end of 2018 (Figure 12), and the cumulative investment steadily increased during the following 27 years. The cumulative afforestation area accounted for 65.77% before 2000 but only 34.23% in the next 19 years. Hence, most afforestation measures occurred in the 1990s, and the SCSs did not immediately exhibit a corresponding increase. It has been proved that there was a nonlinear relationship between the EE and SCSs, and there was usually a lag phase during the early implementation of the EE projects [49]. The afforestation area of the APTM has already achieved the expected goal for 2050 (3.56 million hectares), and economic development and soil protection should remain balanced. The GGP should continue to be implemented in critical areas, and national policies should take into account the amount of increase in the proportion of afforestation.

Land use changes mainly impacted the SCSs through their different vegetation cover types [53]. Due to the implementation of a series of EE projects, the land use pattern of the Taihang Mountain area changed dramatically from 1980 to 2020. The areas of forest and construction land increased in general, while the areas of farmland and grassland decreased, especially in the hilly zone (Table 4). The construction land had the largest growth rate of all of the land use types, which was manifested as a sharp increase in the sub-alpine zone. Therefore, the SCSs in these high-altitude regions may face serious threats. In the hilly zone, many farmland areas were converted to forest land, probably because of the policy of the GGP. The rate of decrease of the farmland gradually decreased with increasing altitude, and the GGP was mainly implemented in the hilly zone. Therefore, the EE measures had the most excellent effect in this zone. That is, the implementation of the GGP reduced agricultural production in specific regions and effectively enhanced the SCSs of the land surface [54]. The evaluation of the EE and its effect on the SCSs will be emphasized and its difficulty will be discussed in the future.

### 4.2. Spatial and Temporal Variations in the Soil Conservation Services in Mountainous Regions

The conclusions of different studies vary greatly. Many previous studies have reported an increasing trend in SCSs, such as in the Loess Plateau region of China, where the average SC increased significantly from 2000 to 2018 [55]. Several researchers have reported that SCSs decreased in their study areas from 2000 to 2015 [56], while other researchers clarified remarkable increasing trends during 1990–2000 and decreasing trends during 2000–2010 [23]. In this study, the amount of SC increased from 1980 to 2020 on the whole, while it exhibited a decreasing trend during part of the study period (2000–2010), which is consistent with the results of Fu [12]. Therefore, the quantification of the SCSs should be calculated in a specific time period.

Spatial heterogeneity was an essential characteristic of the SCSs. The vital SC function was closely related to the terrain conditions and the vegetation structure. It was influenced by factors such as the regional climate, slope, soil type, and vegetation cover. In this study, a key reason was that forestland occupied a more significant proportion in the high-altitude areas, resulting in a richer and more complex ecosystem, which led to a higher soil conservation capacity. This result is consistent with those of Fang et al. [23]. Overall, the spatial and temporal characteristics continuously changed [57], possibly due to changes in the ecological elements caused by the implementation of EE.

### 4.3. Influencing Factors of Soil Conservation Services

The SCSs in the Taihang Mountain area exhibited strong spatial variations, which may have been due to the larger variations in the vegetation coverage [29]. The increase in the vegetation cover and the change in its pattern inevitably led to a change in the spatial pattern of the SCSs [58]. Several studies have shown that the driving force of the vegetation coverage on the SCSs was weak in the early stage of EE [59], and the SC function was significantly improved when the vegetation coverage was greater than 60% [60]. The SCSs varied with the vegetation cover [61]. The vegetation cover factor gradually became the dominant driver of the spatial variations in the SCSs after the implementation of the GGP [58,62]. In this study, there were no significant differences on the average driving force of the vegetation cover (NDVI) in the different altitude zones, which was probably likely because the vegetation recovery markedly improved overall as a result of EE.

In addition to the vegetation coverage, the spatial variations in the SCSs were also affected by the land use types. The implementation of EE resulted in land use changes in the Nile River Basin, and land use made the largest contribution to the SCSs [63]. Forests and grasslands had a threshold effect in controlling soil erosion, and forests were shown to have a relatively higher efficacy in terms of SCSs than grasslands [24]. A similar conclusion was reached in this study (Figure 8). In contrast, the spatial variations in the SCSs in the Taihang Mountain area may have been more strongly influenced by farmland. The farmland accounted for the highest proportion of the study area, and it frequently changed as a result of the GGP [29]. Another similar study conducted in the dry valley region confirmed that the land use type significantly affected the water and soil loss processes [63].

The SC function was also influenced by the slope [58], which was one of the most crucial terrain factors. The slope factors exhibited substantial spatial variations in the study area [38] and were considered the dominant factor controlling the spatial characteristics of the SCSs [64,65]. The primary emphasis of the GGP policy was the transformation of sloping cropland (particularly cropland on steep slopes, i.e., >25°) into forest land or grassland [66]. The slope is a comprehensive index, and the implementation of the GGP may have weakened the impact of the slope factors on the SCSs, especially in the hilly zone. In this study, the influence of the slope on the SCSs was the greatest on the whole. This conclusion is similar to those of many other studies [67,68].

### 4.4. Limitations of This Study and Future Perspectives

Some limitations remain in understanding the identification of and temporally changes in the dynamics of the drivers of SCSs. First, the uncertainties of the identification of the drivers may include the factors not considered, for instance, the effect of the net primary productivity (NPP) [69] and the age of the ecosystem [70]. We selected only the main factors that affect SCSs. Thus, the results may have regional limitations. Second, we analyzed the influencing factors by GDM model. Although the independent variables were divided into more categories as much as possible, chasing for more accurate results. It was not the best analysis method for losing some valuable information. Third, due to the absence of some meteorological data, the estimated R may inevitably differ from the actual conditions, even though it was calculated using the most suitable formula, i.e., the ANUSPLIN interpolation method. Fourth, the scale effect was worth discussing. Data sources and analysis results were based on the Taihang Mountain area, so the results may only be applicable to this study area. We analyzed changes in the SCSs over a relatively long period. There were only five years of data in this study, and some resolutions were not high (i.e., the LULC data). The spatial heterogeneity of SCSs may be more accurately distinguished if the resolutions were improved in our subsequent studies.

We propose three perspectives based on the results of this study. First, compared with the GDM model, a well-accepted random forest algorithm and structural equation model are better methods for identifying the relationships between environmental variables and target factors. Second, recently, some national policies of EE projects have changed significantly. For instance, the arrangement of the new tasks of the GGP was suspended in November 2022, so future studies should assess the changes in and influence mechanism of the SCSs, thus providing a reference for terrestrial SC management. Additionally, more consideration should be given to human activities and EE policies. Third, EE could increase the SC capacity, resulting in changes in the spatial distribution of the SC. There are several uncertainties regarding climate change, such as the increasing frequency of extreme climate events. It is still unknown if the SC capacity in mountainous areas will be resilient to these changing climate conditions. Strengthening the function of the SC is the premise of ecosystem health and sustainable human development.

## 5. Conclusions

This study evaluated the change of SCSs in the Taihang Mountain area based on the InVEST model, thereby improving the accuracy of SC calculations. The average growth rate of the SC was 20.7% higher in the EE implementation areas than in the entire study area. Therefore, EE had a positive effect on SCSs in the past 41 years. The SC exhibited strong spatial heterogeneity in the study area. The distribution pattern of the SCSs was the result of multiple factors. Quantitative identification of SCSs in different altitude zones revealed the differential relationship between SC and its influencing factors. The results indicated that we should adopt different strategies to improve SCSs in mountainous areas. In the hilly zone, EE measures should be strengthened, especially paying more attention to regions with severe soil erosion since IEE dominated the distribution of the SCSs, with the most considerable explanatory power of 34.63%. In the sub-alpine zone, where SC is mainly influenced by slope and NDVI, land managers must protect ecological land, particularly in steep terrain. The Interaction detector analysis of GDM shows that we should consider the comprehensive interactions of multiple factors according to local conditions.

## Figures and Tables

**Figure 1 ijerph-20-03427-f001:**
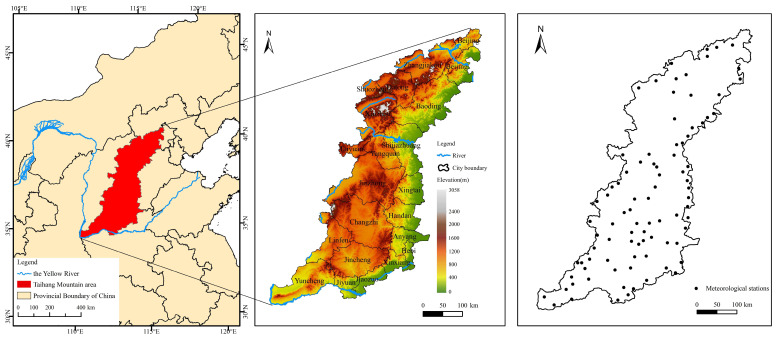
Locations of the Taihang Mountain study area and the meteorological stations.

**Figure 2 ijerph-20-03427-f002:**
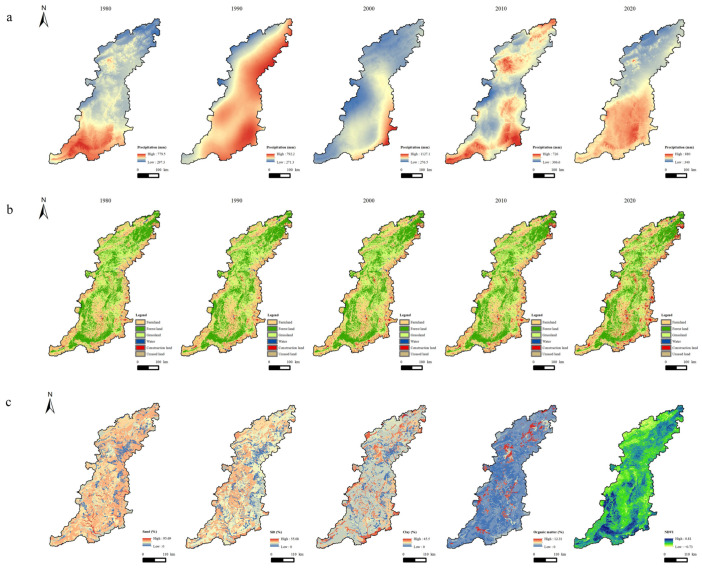
Data information used in this study. (**a**) Precipitation, (**b**) LULC from 1980 to 2020, and (**c**) soil texture and NDVI.

**Figure 3 ijerph-20-03427-f003:**
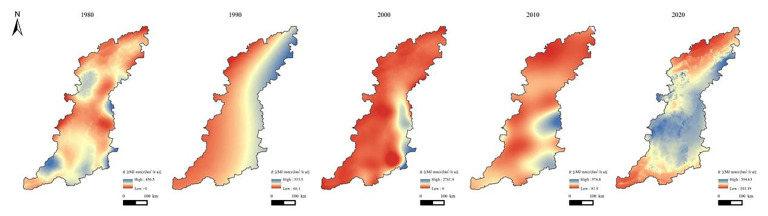
Spatial distribution of rainfall erosion factor (*R*) in the Taihang Mountain area from 1980 to 2020.

**Figure 4 ijerph-20-03427-f004:**
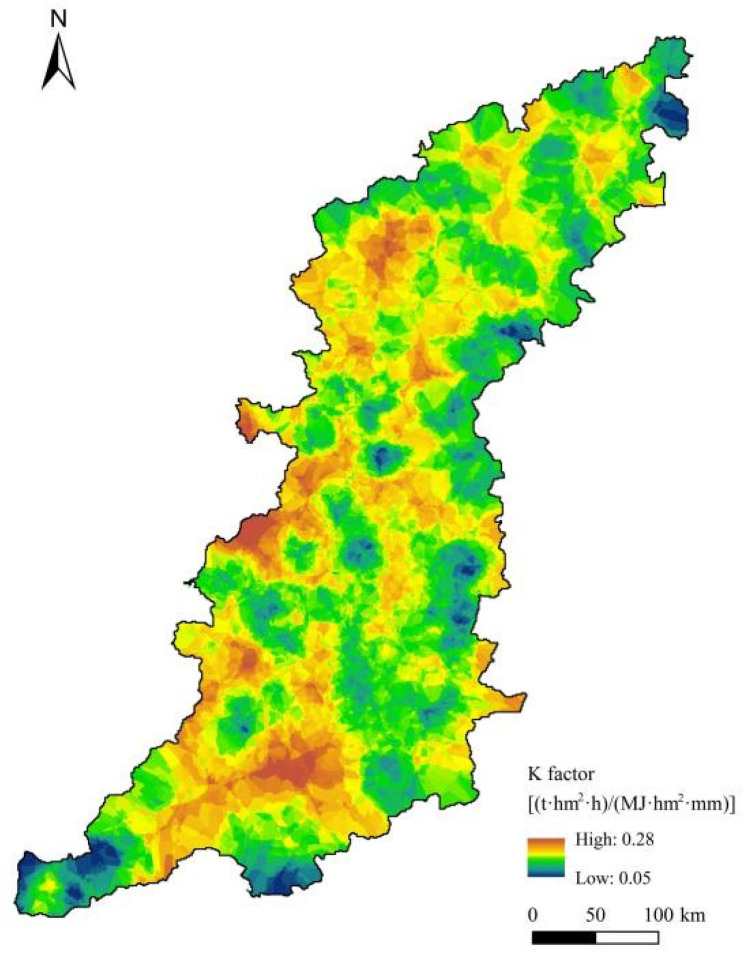
Spatial distribution of soil erodibility factor (*K*) in the Taihang Mountain area.

**Figure 5 ijerph-20-03427-f005:**
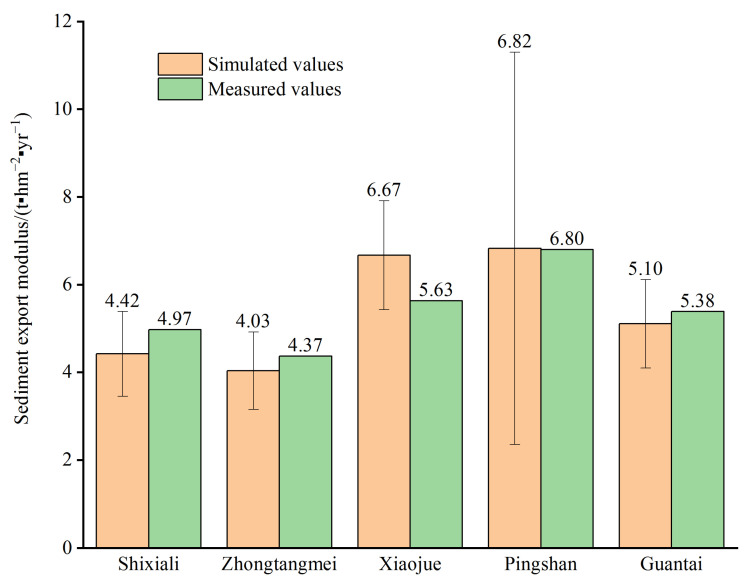
Results of model validation in different basins.

**Figure 6 ijerph-20-03427-f006:**
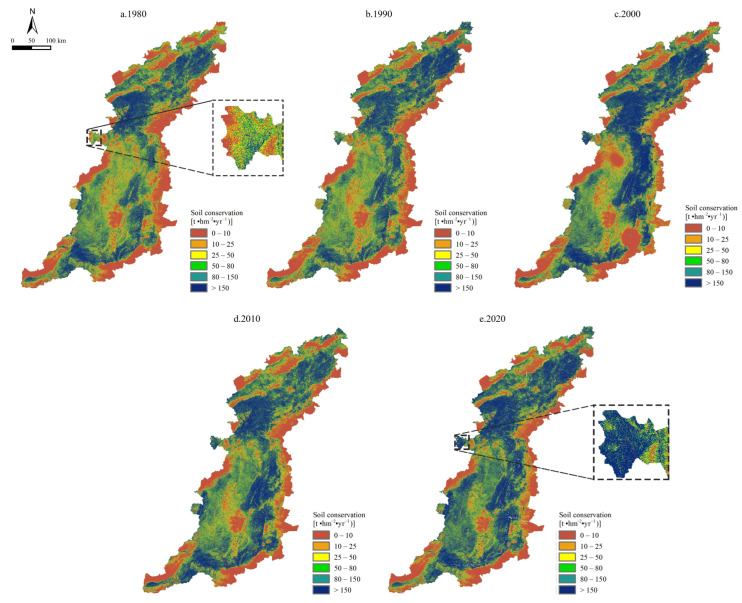
Spatial distribution of SCSs in the Taihang Mountain area from 1980 to 2020.

**Figure 7 ijerph-20-03427-f007:**
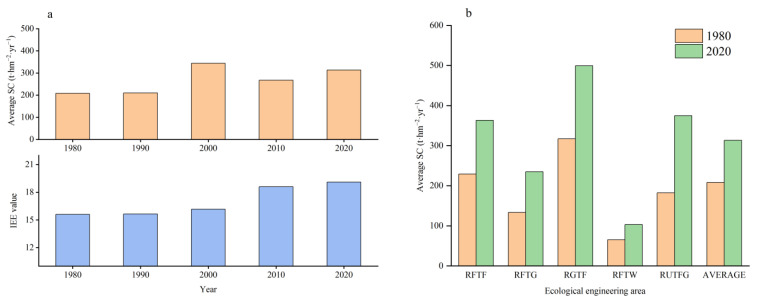
(**a**) Intensity of ecological engineering (IEE) and average SC from 1980 to 2020 in the Taihang Mountain area. (**b**) Differences in the average SC values in the different ecological engineering areas in 1980 and 2020. RFTF—returning farmland to forestland, RFTG—returning farmland to grassland, RGTF—returning grassland to forestland, RFTW—returning farmland to wetland, RUTFG—returning unused land to forestland and grassland.

**Figure 8 ijerph-20-03427-f008:**
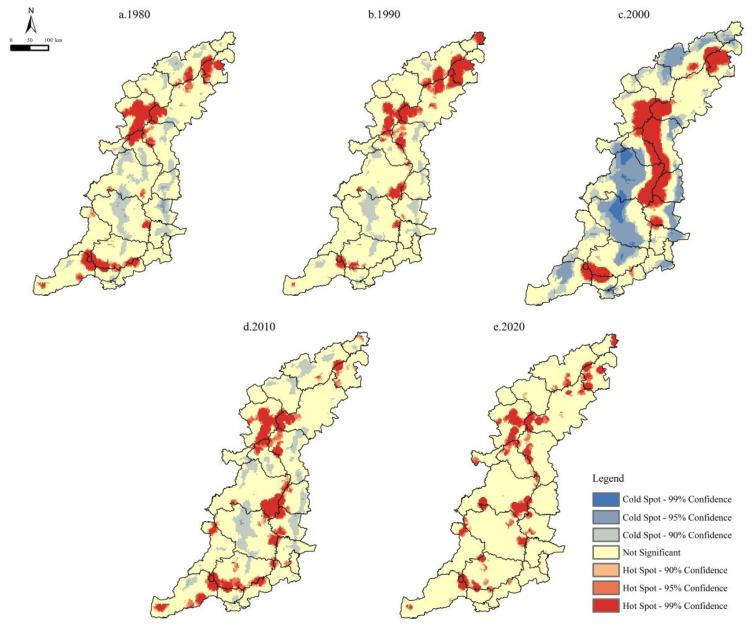
Spatial distributions of SCS hot and cold spots.

**Figure 9 ijerph-20-03427-f009:**
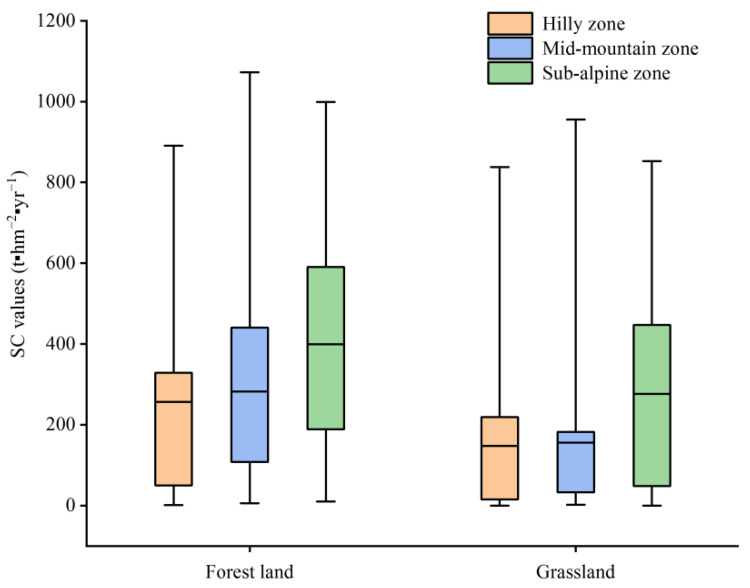
Bar chart of the SC value in the different altitude intervals in 2020.

**Figure 10 ijerph-20-03427-f010:**
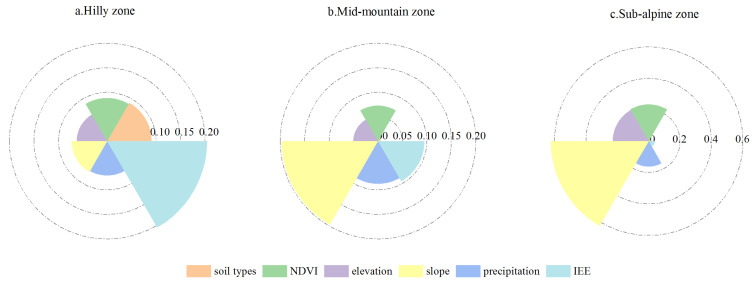
Q values of factors influencing the SCSs in the different elevation zones in 2020.

**Figure 11 ijerph-20-03427-f011:**
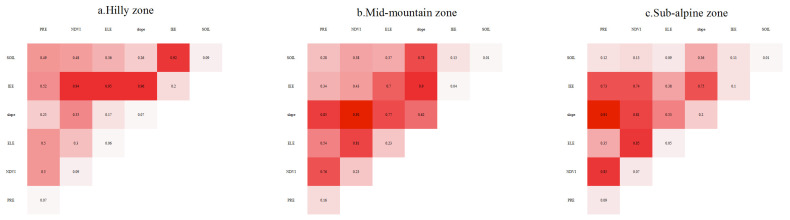
Statistics of q values of interaction factors in the different elevation zones.

**Figure 12 ijerph-20-03427-f012:**
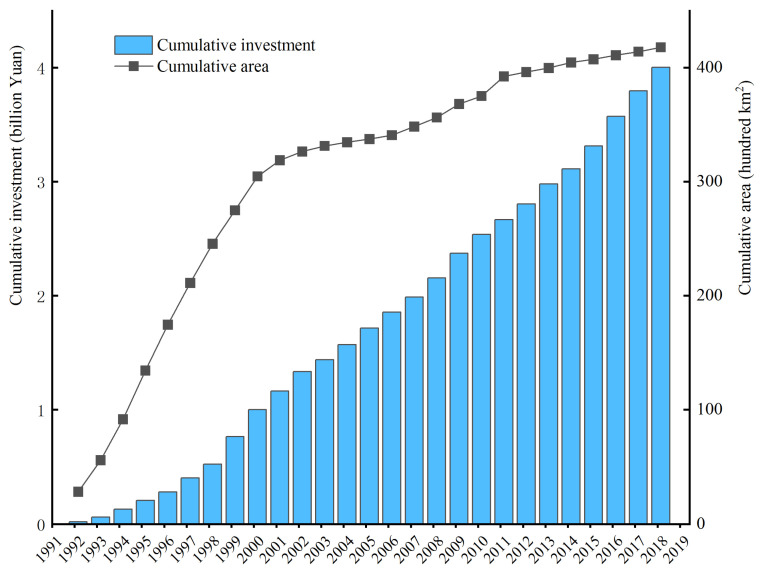
Cumulative area and cumulative investment of APTM from 1992 to 2018.

**Table 1 ijerph-20-03427-t001:** Datasets and resources used in this study.

Data Types	Contents	Resolution or Spatial Distribution	Data Sources
Topographic data	DEM	30 m	http://www.gscloud.cn/ (accessed on 13 October 2020).
Meteorological data	Monthly precipitation data	88 points	Meteorological administrations in the study area
Soil data	Percentage composition of sand, silt, clay, and organic matter	1:1,000,000	China’s second national soil survey
Vegetation data	Monthly NDVI data for 2020	1000 m	https://search.earthdata.nasa.gov (accessed on 9 September 2021).
LULC	Farmland, forest land, grassland, construction land, water, unused land	1000 m	https://www.resdc.cn/ (accessed on 21 May 2021).

**Table 2 ijerph-20-03427-t002:** Proportions of cold spots, hot spots, and non-significant regions from 1980 to 2020.

Types	1980 (%)	1990 (%)	2000 (%)	2010 (%)	2020 (%)
Cold spots	12.54	7.42	31.94	10.76	0.03
Not significant	72.71	77.70	49.57	72.56	86.20
Hot spots	14.75	14.88	18.49	16.68	13.77

**Table 3 ijerph-20-03427-t003:** Average SC and proportions of forest land and grassland in different altitude zones.

Altitude Interval	Forest Land (%)	Grassland (%)	Average SC (t hm^−2^ yr^−1^)
Hilly zone	9.13	20.41	149.71
Mid-mountain zone	33.54	27.48	359.56
Sub-alpine zone	57.59	34.19	456.96

**Table 4 ijerph-20-03427-t004:** Land use changes in the different geomorphic zones from 1980 to 2020.

Land Use Types	Categories	Hilly Zone	Mid-Mountain Zone	Sub-Alpine Zone
Farmland	Variation (km^2^)	−2918	−1810	6
	Rate (%)	−13.96	−5.84	0.73
Forest land	Variation (km^2^)	291	−31	−135
	Rate (%)	9.85	−0.1	−1.92
Grassland	Variation (km^2^)	−368	−1006	−50
	Rate (%)	−4.83	−3.97	−1.2
Water	Variation (km^2^)	−87	−61	5
	Rate (%)	−6.97	−7.88	40.7
Construction land	Variation (km^2^)	2897	2657	114
	Rate (%)	100.1	140.76	765.35
Unused land	Variation (km^2^)	44	−69	−2
	Rate (%)	81.07	−54.63	−27.72

## Data Availability

The datasets analyzed during the current study are available from the corresponding author on reasonable request.

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
