# Peer review of "Spatial-Temporal Variations in of Soil Conservation Service and Its Influencing Factors under the Background of Ecological Engineering in the Taihang Mountain Area, China"

_ijerph, 2023, doi:10.3390/ijerph20043427_

Round 1

Reviewer 1 Report

A brief summary

This article is interesting, although its conclusions are the expected. The identification of slope as the most important factor in soil conservation in mid-mountain and sub-alpine zones is not surprising. However, the quantification of the influence (ranking) of other factors on soil conservation is more interesting (Fig. 9). Perhaps, its greatest contribution is the combination of special and temporal analysis, so that changes and trends can be showed and analysed over time in the study area. It is a paper with a technical profile that does not represent a significant advance in terms of basic research, since it applies already known methodologies. However, it does offer an interesting and solid analysis as a case study. Therefore, it is considered interesting enough for publication after reviewing a series of issues that will be commented after.

Although methodologically the study is solid, the quality of the text is sometimes low. In general, it is possible to understand what the authors wanted to say, but there are some confusing sentences that cannot fully understood (e.g. lines 17-18, 43-44, 107-109, etc.). For this reason, I recommend a professional edition of the text, since it would contribute to significantly improve the final result. Some sections are better written than others, but the entire text could be improved if it is reviewed by a professional editor. Other repetitive problem is that some commas should be full stops (e.g. lines 14, 34, 41, etc.).

Finally, I detail a few minor problems in the next section.

Specific Comments

Lines 30-31 (Keywords): search engines use words present in the title, abstract and this section. If you are repeating words used in the title at the keywords, you are losing visibility. I suggest change three of the five current keywords: Taihang Mountain Area; Soil conservation; Ecologica Engineering. You could use for example: NDVI; slope; soil types.

Line 110 (Materials and Methods): please, define this acronym (LULC = Land Use and Land Cover), as you do with NDVI in line 115.

Lines 175-177 (Materials and Methods): IC0 and Kb are calibrations parameters for SDR (Sediment Delivery Rate. But mention them here has no sense. Why are you doing that? You do not show how to calculate the SDRi. Therefore, this paragraph doesn’t explain something showed in the paper. You should revise it.

Line 189 (Materials and Methods): it would be better if you use here “Quantification of the intensity of ecological engineering (IEE)” instead of “Quantification of ecological engineering intensity”. You have to define all the acronyms the first time you use them.

Line 202 (Materials and Methods): add the reference for Xu Yong et al., i.e. [49].

Lines 218-223 (Materials and Methods): please remove the bold style here.

Line 239 (Results): you are here using some acronyms. Quote the caption in Fig. 6 for a complete description of these acronyms.

Line 314 (Results): In Fig. 9, use the same titles than in Fig 10, i.e. a. Hilly zones, b. Mid-mountain zone, c. Subalpine zone.

Line 330 (Results): please, capitalize the beginning of each zone name.

Line 419 (Discussion): NPP = Net Primary Production. Please, specify it.

Line 433 (Conclusion): correct the typo (exist instead of exsit).

References: please, review the style because there are some works that do not follow the general guidelines: some years of publication are in different order within the reference regarding other works, pagination data is sometimes absent, some acronyms are used for journal titles and other times the complete journal title is cited, etc.

Reviewer 2 Report

The article discusses some interesting findings. There are some types in the manuscript which need to be corrected. Also, some figures were not cited in the text. Following are some specific comments.

Title: Add the country name at the end. If not, international readers cannot understand the location of the study. To me, the title is lengthy. If possible, it is better to shorten the title.

There are some typos: Line 34-35: spacing and a capital letter after the comma, line 144

Introduction: It is important to link soil conservation to the UN Sustainable Development Goals.  

Lines 44-48: The sentence is lengthy and difficult to understand. Better rewrite it.

Line 104: Better to add a sentence before the bullet points. Something like “The following data was used …..”

2.3. InVEST model and its validation: Please add some information about the model. It was not described properly.

Line 144: The sentence should start with a capital letter

Line 144: The number of the reference should be placed after the author’s name (applied by Ma [44])

Line 159: Add the original citation of the EPIC model and define the abbreviation first.

Line 160: No need to mention “research” after the author’s name

Line 180: There is an issue with the units. Please check.

Line 182-183: t/hm2â–ªa. What does “â–ª” mean?

Figure 4. Results of model validation in different basins: should go to the results section. Not methodology.

Line 202: Add reference number

Lines 218-223: Why bold?

Line 227: Units. What does “â–ª” mean?

Figure 6b: Better to add error bars since the values are averages

Line 335-337: Not a complete sentence.

Discussion: It is worth linking the results to future climatic trends. For example, what may be happened in the future under current climatic trends?

Conclusions: Instead of points, combine to one paragraph.

Reviewer 3 Report

This manuscript (ijerph-2068781) tries to analyze the spatial-temporal variation and its influencing factors of soil conservation service under the background of ecological engineering in Taihang Mountain area, China by using the Invest model and GeoDetector method. It is a complete manuscript and the amount of the work is enough. That being said, the authors need to highlight the novelty and contribution of their research as compared with previous research. My detailed comments and suggestions are presented as follows:

(1) Most contents in the Abstract are related to this study area only, which lacks broader international implications. In addition, the scientific question or research gap is missing in the Abstract.

(2) Similarly, the Introduction Section is not very strong because the authors failed to raise an important scientific question beyond this study area. Therefore, potential readers can hardly identify the need that the authors should have to provide a new solution from an international perspective (see below for example).

Modelling surface runoff using the soil conservation service-curve number method in a drought prone agro-ecological zone in Rwanda. International Soil and Water Conservation Research, 2019, 9-17

(3) Figure 1: the meteorological stations are distributed unevenly in the whole study area, which may affect the analysis in this study area.

(4) Section 2.2. Data: it is suggested that the data information can be presented in a new table and a figure.

(5) In addition, the authors need to provide the specific details of the different datasets, such as the pre-processing processes, year, and accuracies.

(6) Figure 2. Spatial distribution of rainfall erosion factor (R) in Taihang Mountain area from 1980 to 2020: from this figure, we can see that the dynamic change of R is very huge, why?

(7) I guess the spatial resolution of the Invest modelling is only 1000 meter, is it too coarse for detailed analysis?

(8) Although the GeoDetector is a widely used and helpful method, it should also be used with caution because all the independent variables need to be categorized. Many valuable information will be inevitably lost during the categorization procedure, and there is no best way for the categorization.

(9) Therefore, it is suggested that the machine learning methods, such as the well-accepted random forest algorithm, can be considered in future studies (see below for discussions). These kinds of methods have been proved to be effective for analyzing the relationships between human activities and environment problems.

Measuring the relationship between morphological spatial pattern of green space and urban heat island using machine learning methods. Building and Environment, 2023, 228: 109910.

On the Scale Effect of Relationship Identification between Land Surface Temperature and 3D Landscape Pattern: The Application of Random Forest. Remote Sens. 2022, 14(2), 279

(10) Figure 5. Spatial distribution of SCs in Taihang Mountain area from 1980 to 2020: in this figure, it is suggested that the differences between different years should be enlarged to make it clear to the readers.

(11) Since there are only five years (samples) of data, is it meaningful to conduct a linear regression analysis in the Figure 6?

(12) In addition, the authors devote too much space to describing the results of this specific study area in the Discussion Section, which lacks the broader planning implications from an international perspective.

(13) Finally, the authors need to discuss the limitations of this study and the future research.

Round 2

Reviewer 2 Report

The paper has been improved based on the comments. However, there are few minor typos. For example, part of the Conclusions is in bold letters. 

Author Response

We apologize for the language and format problems in the manuscript. The language presentation was improved with assistance from a native English speaker with an appropriate research background. Thank you for pointing out this obvious problem. We have modified it.

Reviewer 3 Report

This is the second time I am reviewing this manuscript (ijerph-2068781), which has been significantly criticized because of the lack of novelty and solid scientific foundation. I would like to remind the authors that "International Journal of Environmental Research and Public Health" is a considerably prestigious journal and thus very competitive to publish. The scientific rationale behind the literature review aimed at publishing such a prestigious journal should be crystal clear and highly novel. I believe the authors could have taken the opportunity of the first revision and significantly improve their work based on the reviewer’s comments and suggestions. Unfortunately, there are still many major concerns remaining in this manuscript, which I believe they were either not explained well or they were somehow misleading. The remaining issues are repeated and summarized below:

(1) I do not agree that the relation between ecological engineering and soil conservation can be deemed as a new and important contribution in this research field.

(2) Most of the contents are still related to this study area only, which still lacks broader international implications. I do not agree that the Taihang Mountain is famous around the world. Scholars in the other countries and regions do not really care about the case study of soil conservation problems in this area.

(3) The meteorological stations are distributed unevenly in the whole study area, which will affect the analysis in this study area. The authors still did not solve this very severe problem. It is very severe that a few counties lacked meteorological data.

(4) Again, Section 2.2. Data: it is suggested that the data information should be presented in a new figure.

(5) Again, the authors need to provide the specific details of the different datasets, in particular the accuracies.

(6) Still, a thorough and criticism-featured Literature Review section is needed. The existing literature review is insufficient.

(7) In the response 8, the authors mentioned that: "In this paper, I divided the independent variables into more categories as much as possible and randomly set up hundreds of sampling points throughout the study area chasing for more accurate results". Unfortunately, dividing the independent variables into more categories as much as possible does not guarantee the best grouping.

(8) The well-accepted random forest algorithm considers the distribution of dependent variables, and it is more reasonable to explain the relationships between environmental variables and target factors. The authors need to explain these limitations of this study and the future research in the Discussion Section.

(9) In the Conclusions Section, the conclusions just simply repeated the results. In addition, what are the advantages and disadvantages of the model/results compared with previous similar studies? These sentences are not conclusions. In other words, the manuscript needs a strong take-home message in order to increase its potential scientific impact.

(10) The scale effect is also worth discussing. What is the difference between the modelling results at different resolutions? What are its impacts on accuracy?

(11) Some grammatical errors, typos can be found in the manuscript. A proof reading by a native English speaker should be conducted to improve both language and organization quality.

Author Response

Thank you very much for your professional comments. ALL the precious comments and advice have been further revised. It is crucial for guiding our future research. Please see the attachment for detailed responses.
